# Sexual Satisfaction among Lesbian and Heterosexual Cisgender Women: A Systematic Review and Meta-Analysis

**DOI:** 10.3390/healthcare11121680

**Published:** 2023-06-07

**Authors:** Ana Macedo, Eunice Capela, Manuela Peixoto

**Affiliations:** 1Faculdade de Medicina e Ciências Biomédicas, Universidade do Algarve, 8005-139 Faro, Portugal; ecapela@sapo.pt; 2Algarve Biomedical Center, 8005-139 Faro, Portugal; 3Centro de Psicologia da Universidade do Porto, Faculdade de Psicologia e de Ciências da Educação, Universidade do Porto, 4099-002 Porto, Portugal; nelinha.peixoto@gmail.com

**Keywords:** sexual and gender minority, sexual satisfaction, lesbian women, meta-analysis

## Abstract

Background: Sexual satisfaction is a complex, multifaceted, and broad concept that is influenced by several factors. The minority stress theory posits that sexual and gender minorities are at a particular risk for stress due to stigma and discrimination at the structural, interpersonal, and individual levels. The aim of this systematic review and meta-analysis was to evaluate and compare the sexual satisfaction between lesbian (LW) and heterosexual (HSW) cisgender women. Methods: A systematic review and meta-analysis were conducted. We searched the PubMed, Scopus, Science Direct, Websci, Proquest, and Wiley online databases from 1 January 2013 to 10 March 2023 to identify the published observational studies on sexual satisfaction in women according to their sexual orientation. The risk of bias in the selected studies was assessed using the JBI critical appraisal checklist for the analytical cross-sectional studies. Results: A total of 11 studies and 44,939 women were included. LW reported having orgasms during a sexual relationship more frequently than HSW, OR = 1.98 (95% CI 1.73, 2.27). In the same direction, the frequency of women reporting “no or rarely” for having orgasms during their sexual relationships was significantly lower in the LW than the HSW, OR = 0.55 (95% CI 0.45, 0.66). The percentage of the LW who reported having sexual intercourse at least once a week was significantly lower than that of the HSW, OR = 0.57 for LW (95% CI 0.49, 0.67). Conclusions: Our review showed that cisgender lesbian women reached orgasm during sexual relations more often than cisgender heterosexual women. These findings have implications for gender and sexual minority health and healthcare optimization.

## 1. Background and Rational

The minority stress theory posits that sexual and gender minorities are at a particular risk for stress due to stigma and discrimination at the structural, interpersonal, and individual levels [1]. This stress, in turn, elevates the risk of adverse health outcomes across several domains, including sexual satisfaction.

Many studies have pointed out that sexual minorities experience significant health disparities [2,3,4]. These differences are associated with a model of intersectionality that includes social, biological, and economic components [5]. Despite this multidimensionality, the differences described focus primarily on mental health [6,7], sexually transmitted infections and diseases, body image changes, and eating disorders [8,9]; however, sexual well-being, despite its importance, is not usually taken into account in these analyses. 

Sexual satisfaction is a complex, multidimensional, and broad concept that includes individual, relational, and contextual dimensions [10] and is conceptualized as a sexual right by the World Health Organization [11] and the World Association for Sexual Health [12,13,14]. According to Meston and Trapnell, 2005 [15], sexual satisfaction in women is described as a multifaceted construct that includes relational dimensions such as compatibility between partners, communication patterns, and the absence of relational concerns, as well as individual dimensions such as feelings of contentment and satisfaction with one’s sex life, and the absence of personal distress and concerns. Although sexual satisfaction is often perceived as the opposite of sexual distress, both constructs appear to have a more complex understanding, with sexual satisfaction being partially independent of sexual distress and being unable to be measured by an absence of sexual distress [16].

According to Pascoal et al. [10], the heterosexual lay definitions of sexual satisfaction include two core dimensions related to sexual well-being at the personal level, and dyadic and relational processes. The individual component includes positive dimensions such as reaching orgasm, experiencing sexual arousal, pleasure, and positive effect, while the dyadic component includes the frequency of sexual activity, intimacy, romantic feelings, and their expression. In contrast, the lay definition of sexual satisfaction among sexual minorities, which includes women in lesbian partnerships, involves these individual and dyadic components, as well as social discourses about sexual orientation [17]. The greatest similarity between the heterosexual and lesbian definitions of sexual satisfaction is positive attitudes toward sexuality and pleasure, rather than a simple absence of sexual difficulties.

Human sexuality is a natural and important part of peoples’ lives and well-being. The underlying interactions affecting sexual satisfaction are complex, and sexual orientation differences partly remain to be identified, as well as explained [18].

Our aim with this systematic review and meta-analysis was to evaluate and compare the sexual satisfaction between lesbian (LW) and heterosexual (HSW) cisgender women.

## 2. Methodology

### 2.1. Study Design

The PRISMA guidelines (Preferred Reporting Items for Systematic Reviews and Meta-Analyses) [19] and recommendations of the Cochrane Handbook were followed [20]. There were no requirements for an ethical review of this work, because no human participants were involved.

### 2.2. Search Strategy

We searched the PubMed, Scopus, Science Direct, Websci, Proquest, and Wiley online databases from 1 January 2013 to 10 March 2023 to identify the published observational studies on sexual satisfaction in women according to their sexual orientation.

The search strategy used in PubMed is described below. The strategy was modified accordingly for its use in the other databases. 

((“homosexuality, female” [MeSH Terms] OR (“homosexuality” [All Fields] AND “female” [All Fields]) OR “female homosexuality” [All Fields] OR “lesbianism” [All Fields] OR “sexual and gender minorities” [MeSH Terms] OR (“sexual” [All Fields] AND “gender” [All Fields] AND “minorities” [All Fields]) OR “sexual and gender minorities” [All Fields] OR “lesbian” [All Fields] OR “lesbians” [All Fields] OR “lgbt*” [All Fields]) AND (“orgasm” [MeSH Terms] OR “orgasm” [All Fields] OR (‘sexual” [All Fields] AND “satisfaction” [All Fields]) OR “sexual satisfaction” [All Fields]) AND (“femal” [All Fields] OR “female” [MeSH Terms] OR “female” [All Fields] OR “females” [All Fields] OR “females” [All Fields] OR “femals” [All Fields] OR (“womans” [All Fields] OR “women” [MeSH Terms] OR “women” [All Fields] OR “woman” [All Fields] OR “womens” [All Fields] OR “womens” [All Fields]) OR (“womans” [All Fields] OR “women” [MeSH Terms] OR “women” [All Fields] OR “woman” [All Fields] OR “womens” [All Fields] OR “womens” [All Fields]))) AND (y_10[Filter])

There were no publication status and language restrictions on selecting the studies.

### 2.3. Inclusion and Exclusion Criteria

The inclusion criteria were: 1. studies of a cross-sectional, observational type (including cross-sectional analyses of longitudinal studies); 2. participants: >18 years old; heterosexual and lesbian cisgender women; 3. comparison group/condition of interest: sexual orientation (heterosexual versus lesbian); 4. primary outcome: sexual satisfaction–orgasm frequency; 5. secondary outcomes: the frequency of sexual intercourse and arousal difficulties; and 6. settings: community. The exclusion criteria were: 1. studies exclusively focused on transgender or nonbinary subjects; 2. global outcomes for lesbian and bisexual subjects; and 3. experimental studies.

### 2.4. Screening and Data Extraction

An initial screening of the titles was conducted by (AM) based on the inclusion criteria. Duplicates and studies clearly not associated with the review objectives were excluded. Then, the abstracts were independently screed by two reviewers (AM and EC) based on the above-established criteria. The relevant studies and those in which the abstract raised doubts had their full texts independently evaluated by the two reviewers.

All disagreements were solved by consensus.

The following data were then extracted from the included studies: the first author, publication year, country, study design, sample size, study population details, outcomes, results, and authors’ main conclusions.

Sexual satisfaction was evaluated through various parameters, which included the frequency with which the women reached orgasm in a sexual relationship, the degree of difficulty in becoming aroused, pain during or after sex, a lack of interest in sex, the frequency of sexual activity, and global sexual satisfaction.

### 2.5. Risk of Bias Assessment

Two reviewers independently assessed the risk of bias in the selected studies using the JBI critical appraisal checklist for analytical cross-sectional studies [21]. This tool used eight criteria to evaluate the overall methodological quality of a study. The criteria included: the sample inclusion criteria; a description of the subjects and settings; a valid and reliable measure of exposure; an objective and standard measure of condition; identifying confounding factors; strategies for dealing with confounding factors; a valid and reliable measure of outcome; and an appropriate statistical analysis. Disagreements were resolved through discussion.

### 2.6. Meta-Analysis

The RevMan 5.4.1 (The Cochrane Collaboration) online software was used. Comparable data from studies with similar populations and outcomes were pooled using forest plots. The odds ratio (OR) with 95% confidence intervals (CIs) for dichotomous data was used as the effect measure. The statistical heterogeneity among the selected studies was measured by using I^2^ in each analysis and a 5% significance level. The random effect model was selected for the analysis carried out, because the true effect sizes underlying all the studies were stochastic.

A sensitivity analysis was conducted to evaluate the robustness of the results by excluding individual studies for each forest plot. 

## 3. Results

### 3.1. Search Results and Study Characteristics

The details of the literature search and screening process can be found in Figure 1. 

A total of eleven studies were selected for this review. The included studies were published between 2014 and 2022. A total of 44,939 women were included, from Australia, Germany, Portugal, Spain, Sweden, the United Kingdom, and the United States. The mean age was 31 years, most participants were white, and more than two thirds had a college degree or had attended a university. The studies’ risk of bias assessment is presented in Table 1. Characteristics of the studies’ populations and studies’ main results are described in Table 2 and Table 3.

In general, the studies recruited their samples online, using several methodologies that included posting ads on social networks and websites specifically targeting people belonging to sexual and gender minorities, and through direct contact with respondents in a snowball approach.

### 3.2. Risk of Bias

The risk of bias in the included studies was low. The major weakness of the studies was in the control of the confounding variables, which was not performed in six of the eleven studies.

### 3.3. Sexual Satisfaction Results

Sexual satisfaction was evaluated in five studies. Although the evaluation methodologies were different, and were therefore not directly comparable, the studies showed no significant differences between LW and HSW in this global assessment.

### 3.4. Frequency of Sexual Intercourse

The frequency of sexual intercourse was presented in four studies involving 5339 women, showing a lower frequency for LW when compared to HSW.

A fixed-effects Mantel–Haenszel (M-H) model meta-analysis showed that the percentage of LW who reported having sexual intercourse at least once a week was significantly lower than that of HSW, OR = 0.57 for LW (95% CI 0.49, 0.67); I^2^ = 65%, (Figure 2).

### 3.5. Orgasm Frequency

Six studies involving 38,760 women reported the frequency of subjects who reported having orgasms always or usually.

A meta-analysis using a fixed-effects Mantel–Haenszel (M-H) model showed that there was a statistically significant difference between LW and HSW. LW reported to have orgasms during sexual relations “always or usually” more frequently than HSW, OR = 1.98 (95% CI 1.73, 2.27); I^2^ = 87%, (Figure 3).

Seven studies involving 39,525 women reported the frequency of subjects who reported not having or rarely having orgasms. In the same direction, a meta-analysis using a fixed-effects model found that the percentage of LW who reported “not having or rarely having orgasms” during sexual relations was significantly lower than that of HSW, OR = 0.55, (95% CI 0.45, 0.66); I^2^ = 83%, (Figure 3).

### 3.6. Arousal Difficulties

Three studies including 9825 participants reported the frequency of arousal difficulties. A meta-analysis with a fixed-effects Mantel–Haenszel (M-H) model showed no statistically significant differences in the percentage of LW with arousal difficulties compared to HSW, OR = 0.79, (95% CI 0.61, 1.01); I^2^ = 0%, (Figure 4).

In all the meta-analyses, a sensitivity analysis showed that the results remained unchanged after the exclusion of each individual study.

## 4. Discussion

Our data included eleven studies and a total of 2294 LW and 42,645 HSW. The results were relatively homogeneous and showed that the percentage of LW who reported to have orgasms during all or most of their sexual relations was almost two times higher than that of HSW. The percentage reporting never or almost never reaching orgasm was higher among HSW. At the same time, LW reported a lower frequency of sexual activity.

In recent decades, the assessment of sexual satisfaction has gained importance and been included in health and well-being indicators. In 2002, the WHO [11], in collaboration with the WAS, highlighted the importance of sexual health, including key factors such as information and sexual pleasure.

Women’s sexuality, particularly heterosexual women’s sexuality, and predictors of sexual satisfaction are well documented in the literature, with an emphasis on distinctive features such as relational, psychological, and biological dimensions, above and beyond sexual function [32,33,34]. While sexual satisfaction is defined differently by many authors, there are also similarities among most studies. The evaluation of the frequency of orgasm is a relatively simple and transversal measure, although it depends on the self-evaluation and perception of each person.

A discrepancy between sexual satisfaction levels and the frequency of sexual activity was found [35], suggesting that women’s sexual satisfaction is multidetermined. Distinctive factors were identified as promoters of sexual satisfaction, such as communication patterns, the quality of a relationship, and sexual compatibility, whereas other factors were described as sexual satisfaction attenuators, such as sexual disorders and discrepancies in sexual desire levels [36].

A review published in 2021 [33] analyzed the predictors of sexual satisfaction in heterosexual women and included a total of 204 studies. Of the variables analyzed by the authors, the following were highlighted as being correlated with women’s sexual satisfaction: body image, mental health, physical health, orgasm frequency, relationship satisfaction, sexual communication, sexual desire, sexual frequency, sexual functioning, and sexual practices/variety. The authors of this review noted that the methods used to classify sexual satisfaction were highly heterogeneous across the studies and that there were possibly relevant variables that were only considered in a minority of studies, although they seemed to correlate with female sexual satisfaction, such as: sexual openness or sexual thoughts/fantasies.

Data on the predictors of LW’s sexual satisfaction have been overlooked, considering that most studies of women’s sexual satisfaction and its predictors have been conducted with HSW, leading to a gap in the literature on understanding LW’s sexual satisfaction.

A study about midlife women [37] revealed that their sexual satisfaction was partially determined by sexual function and enhanced relational, psychological, and biological dimensions, such as relationship adjustment, psychological well-being, and menopausal symptoms, but was not determined by sexual orientation [38].

In another study [39] that evaluated HSW, LW, and bisexual women, an absence of depressive symptoms, satisfaction in an intimate relationship, better sexual functioning levels, and perceived social support positively predicted sexual satisfaction. Among LW, experiencing internalized homonegativity was also a significant and negative predictor of their sexual satisfaction. A more recent study found evidence that internalized homonegativity does not contribute to lesbian women’s sexual satisfaction, whereas identity pride positively contributes to lesbian women’s sexual satisfaction [40].

The results of our review are in accordance with several studies that have indicated that LW tend to report higher levels of sexual functioning compared to HSW [41,42,43], and more specifically, that LW report better orgasmic function.

In a study on Dutch college women, LW were also much more likely to experience orgasms during sexual activity with a female partner than HSW [44]. In this study, the likelihood of orgasm was strongly related to receiving glans clitoral stimulation: lesbian women were less likely to receive vaginal stimulation during lovemaking than women in mixed-gender relationships, but those who did were significantly more likely to experience orgasm from it [45]. 

Possible justifications for the current findings may be related to the different sexual stimulation observed in LW and HSW and the possibility that LW engage in sexual activity and stimulation that facilitate reaching orgasm. 

One controversial question about LW’s sexual lives is whether they have sexual relations less frequently than HSW [46]. In our review, the frequency of sexual intercourse, evaluated in four studies involving 5339 women, showed a statistically significant difference between lesbian and heterosexual women, with an odds ratio of 0.57 for LW, meaning that the percentage who reported to have sexual intercourse more than once a week was almost half of that for HSW.

One of the problems related to the estimation of sexual satisfaction is that, often, the frequency of sexual intercourse is only considered. Compared to HSW, LW were more likely to usually receive oral sex, to use sex toys, to have sex for longer than 30 min, and engage in gentle kissing, having a broader set of behaviors included their definition of having sex [47,48,49,50]. 

This review has some limitations that should be taken into consideration. First, there may be publications that were not indexed on the databases searched, and some relevant articles may not have been found. Second, as previously mentioned, the definition of sexual activity and satisfaction could be heterogeneous, as well as the study’s methodology.

## 5. Conclusions

In conclusion, our review showed that LW achieved orgasm during a sexual interaction more often than HSW. At the same time, LW reported less frequent sexual intercourse when compared to HSW. The percentage of women reporting arousal difficulties did not differ according to their sexual orientation.

It is critical to understand that sexual satisfaction differs according to sexual orientation and to try to understand which factors are associated with a higher satisfaction, especially in minority groups that have been overlooked compared to cisgender, heterosexual people.

These findings have implications for gender and sexual minority health and healthcare optimization.

## Figures and Tables

**Figure 1 healthcare-11-01680-f001:**
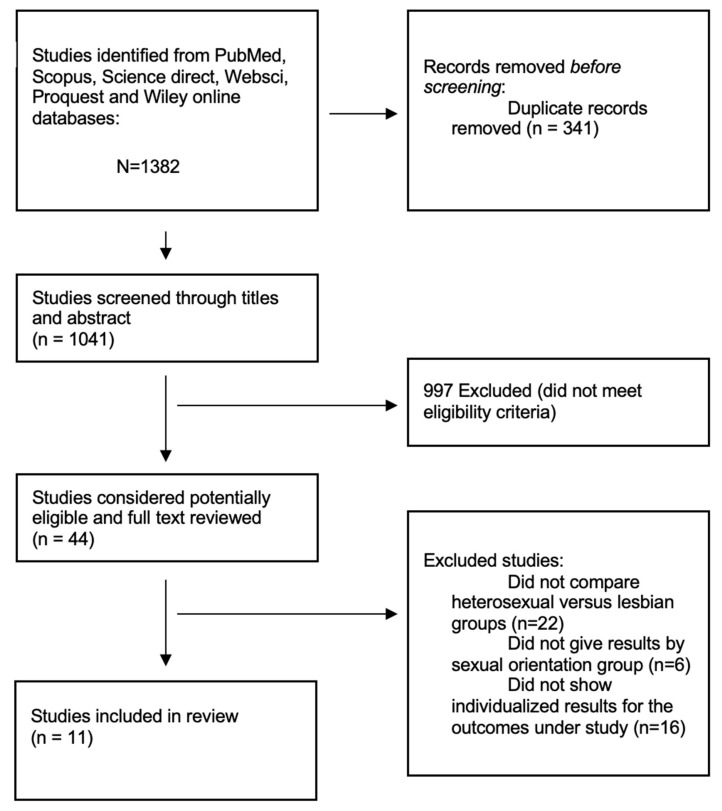
Literature search and screening process—PRISMA flowchart.

**Figure 2 healthcare-11-01680-f002:**
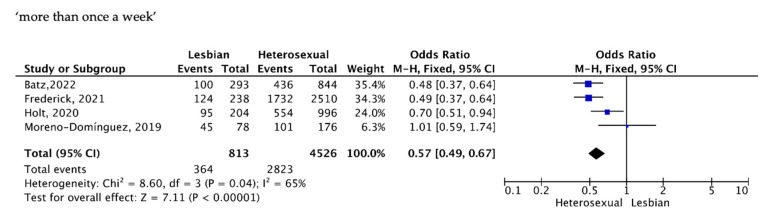
Frequency of sexual intercourse [23,26,28,30].

**Figure 3 healthcare-11-01680-f003:**
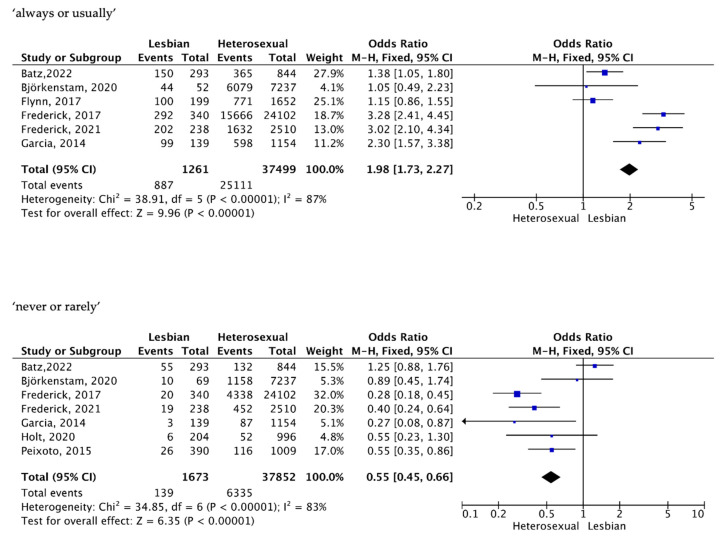
Orgasm frequency [18,23,24,25,26,27,28,31].

**Figure 4 healthcare-11-01680-f004:**
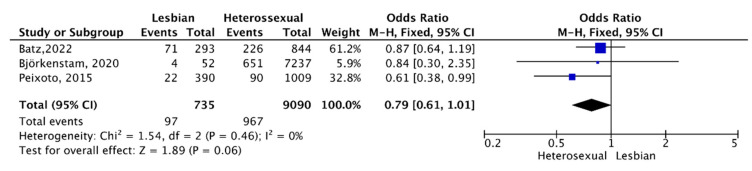
Arousal difficulties [18,23,31].

**Table 1 healthcare-11-01680-t001:** Included studies’ risk of bias assessment.

References	Inclusion Criteria	Subjects and Setting Detailed Description	Valid and Reliable Measure of Exposure	Objective and Standard Measure of Condition	Confounding Factors Identified	Strategies for Dealing with Confounding Factors	Valid and Reliable Measure of Outcome	Appropriate Statistical Analysis
Amos, 2015 [22]	Y	Y	Y	Y	Y	Y	Y	Y
Batz, 2022 [23]	Y	Y	Y	Y	Y	N	Y	Y
Björkenstam, 2020 [18]	Y	Y	Y	Y	Y	N	Y	Y
Flynn, 2017 [24]	Y	Y	Y	Y	Y	N	Y	Y
Frederick, 2017 [25]	Y	Y	Y	Y	Y	Y	Y	Y
Frederick, 2021 [26]	Y	Y	Y	Y	Y	Y	Y	Y
Garcia, 2014 [27]	Y	Y	Y	Y	N	N	Y	Y
Holt, 2020 [28]	Y	Y	Y	Y	Y	Y	Y	Y
Joyner, 2020 [29]	Y	Y	Y	Y	Y	Y	Y	Y
Moreno-Domínguez, 2019 [30]	Y	Y	Y	Y	N	N	Y	Y
Peixoto, 2015 [31]	Y	Y	Y	Y	N	N	Y	Y

Y—Yes: N—No.

**Table 2 healthcare-11-01680-t002:** Studies’ design and population descriptions.

Author	Study Design and Participants	Country	Age Global	Heterosexual Women		Lesbian Women	
				N (Age)	% White/ % University Frequency	N (Age)	% White/ % University Frequency
Amos, 2015 [22]	Participants included men and women who self-identified as heterosexual, gay, lesbian, or bisexual who lived in Australia, North America, or the UK. Were recruited online via posts to various sub-categories of a social news and entertainment-sharing website, where registered users can submit content to share with others. The posts asked for volunteers between 18 and 65 years old to participate in a survey on sexiness and provided a link to an online questionnaire. Participation was voluntary and no incentive was provided.	Australia, North America, UK	22.8 y	468 (--)	--/89%	246 (--)	--/87%
Batz, 2022 [23]	Anonymous nationwide online survey. To spread the questionnaire online, invitations with a link to access the survey were shared via e-mail distribution lists and on social communication networks such as Facebook, Instagram, Twitter, and WhatsApp. Participants were invited to forward the link of the survey (snowball sampling). Cross-sectional study design was carried out during the first confinement in Germany from 20 April to 20 July 2020. Inclusion criteria were a minimum age of 18 years and German language skills. Data collection in 2020.	Germany	--	844 (26–35 y)	--/53%	293 (26–35 y)	--/44%
Björkenstam, 2020 [18]	Data from sexual and reproductive health and rights (SRHR) 2017, based on a Swedish national sample of women and men aged between 16 and 84 years. The paper questionnaires were mailed and the respondents also received an information letter on the survey and its purpose. The respondents were also informed that the questionnaire would be supplemented with register data and that participation was voluntary and anonymous. Data collection in 2017.	Sweden	--	7237 (45–64 y)	--/52%	69 (30–44 y)	--/52%
Flynn, 2017 [24]	Cross-sectional surveys were administered by KnowledgePanel^®^ (GfK), an online panel that uses address-based probability sampling and is representative of the civilian, noninstitutionalized US population. Eligibility criteria for both samples included age of 18 years or older and ability to read English. Data collection in 2013–2014.	US	41 y	1652 (--)	--/--	199 (--)	--/--
Frederick, 2017 [25]	Based on secondary analyses of anonymous data collected via a 172 online survey posted on the official website of NBC News for ten days; 18–65 years. Data collection in 2006.	US	--	24,102 (33.8 y)	84%/47%	340 (36.5 y)	84%/92%
Frederick, 2021 [26]	Secondary analyses of anonymous data collected via an online survey posted on the official news website of NBC News (then called msnbc.com) for ten days. Aged 18 to 65 years; identified as heterosexual or lesbian; indicated they were either dating/seeing only one person, cohabiting, married, or remarried; and were sexually intimate with their partner during the last month. Data collection in 2006.	US	--	2510 (35.6 y)	84%/93%	238 (35.5 y)	83%/93%
Garcia, 2014 [27]	Online questionnaire of single men and women in the US, internet research panels for population-based cross-sectional surveys. Nationally representative research panels compiled based on demographic distributions reflected in the most recent Current Population Survey. Inclusion criteria required being at least 21 years of age and identifying current relationship status as single. Data collection in 2011.	US	40–44 y	1154 (--)	--/--	139 (--)	--/--
Holt, 2020 [28]	Participants had to be at least 18 years and currently in a sexual relationship to be eligible for this study. They were recruited online via posts on the Kinsey Confidential website, Facebook, websites and forums for lesbian and bisexual women, and the American Psychological Association listservs of Society for the Psychology of Women and Society for the Psychological Study of Lesbian, Gay, Bisexual, and Transgender Issues. In order to recruit a more balanced sample, special effort was exerted to recruit conservative and religious women. Participants were offered an opportunity to enter to win one of four $25 Amazon gift cards.	US	--	996 (32.5 y)	90%/98%	204 (34.5 y)	81%/96%
Joyner, 2020 [29]	National Longitudinal Study of Adolescent to Adult Health. By the time of the fourth in-home interview, most respondents were between the ages of 25 and 32 and “currently” in a romantic and/or sexual relationship. Data collection in 2007–2008	US	--	2498 (27.9 y)	60%/67%	98 (27.8 y)	72%/65%
Moreno-Domínguez, 2019 [30]	The women were asked to complete a survey prompted by online advertisements on different websites. Postings were added to general community forums and websites of interest to lesbian and bisexual women in Spain. Participation was voluntary and anonymous; no compensation was offered to survey respondents. 18–62 years. Data collection in 2018.	Spain	25.4 y	176 (--)	--/--	78 (--)	--/--
Peixoto, 2015 [31]	General population who completed an online survey about female sexual problems. The online survey was publicized on several Portuguese LGBT forums, websites, and social networks (focused recruitment). Additionally, an invitation by e-mail was sent via Portuguese universities’ mailing lists. No monetary compensation was given. Data collection in 2012–2013.	Portugal	--	1009 (25.7 y)	67%/--	390 26.3 y	--/59%

**Table 3 healthcare-11-01680-t003:** Studies results and conclusions.

Author	Outcomes	Results	Authors Conclusion	
Amos, 2015 [22]	Multidimensional Sexual Self-Concept Questionnaire (MSSCQ)	Sexual satisfaction—mean (sd): HSW 2.7 (1.2) LW 1.9 (1.3); Sexual attractiveness—mean (sd): HSW 31.3 (7.5) LW 29.0 (8.3)	HSW rated their sexual attractiveness more positively than LW. HSW reported a greater level of sexual esteem and sexual satisfaction and a higher frequency of sexual activity than LW.	
Batz, 2022 [23]	Frequency of masturbation, frequency of sexual intercourse, sexual arousal, capability to enjoy sexual intercourse, and general satisfaction with sexual life	Frequency of masturbation—mean (sd): HSW 2.4 (1.0) LW 2.5 (1.1); frequency of sexual intercourse—“more that once a week”: HSW 51.6% LW 34.1%; sexual arousal—”easy/very easy” HSW 73.2% L 75.7%; capability to enjoy sexual intercourse—“always”: HSW 43.2% LW 51.2%; capability to enjoy sexual intercourse—“never/occasionally”: HSW 15.6% LW 18.7%; general satisfaction with sexual life—“reasonable/full”: HSW 57.9% L 56.5%	Levels of sexual health were lower among HSW compared to LW.	
Björkenstam, 2020 [18]	Lack of interest in sex, felt no pleasure, pain during or after sex, lack of sexual arousal, and no orgasm	Lack of interest in sex: HSW 34% L 37%; felt no pleasure: HSW 9% LW 8%; pain during or after sex: HSW 11% LW 11%; lack of sexual arousal: HSW 9% LW 8%; and no orgasm: HSW 16% LW 15%	LW seemed to have a lower risk for many sexual problems (however not significant). Furthermore, LW had a 7-fold higher risk of experiencing premature orgasm, compared with HSW. Tendencies for lower risk of no orgasm for LW than for HSW. LW seem to be more satisfied. A strong contributor to sexual satisfaction is orgasm ability.	
Flynn, 2017 [24]	Sexual function and satisfaction past 30 days (PROMIS SexFS v2)—interest in sexual activity, vaginal discomfort with sexual activity, satisfaction with sex life, and orgasm ability	Interest in sexual activity—mean (sd): HSW 46.6 (45.8–47.5); LW 46.2 (44.6–47.7), vaginal discomfort with sexual activity—mean (sd): HSW 49.3 (48.5, 50.0) LW 45.5 (44.2, 46.9), satisfaction with sex life—mean (sd): HSW 49.4 (48.6, 50.2) LW 47.4 (45.8, 49.1), and orgasm ability—mean (sd): H 46.7 (45.7, 47.6) LW 50.3 (48.2, 52.4)	Among women, we did not find differences in satisfaction by sexual orientation. COVID-19 pandemic and resulting social constraints had a particular impact on the sexual health of LW.	
Frederick, 2017 [25]	Orgasm frequency (last month); partner orgasms frequency	Orgasm frequency—“usually always”: HSW 65%, L 86%; orgasm frequency—“never/rarely”: HSW 18% LW 6%; partner orgasm frequency—“usually always”—H 95%; L 87%	LW had three times greater odds than HSW of always experiencing orgasm (OR = 2.98, *p* < 0.001). LW were more likely to orgasm than HSW, even when controlling for important contributors to orgasm frequency that might vary by sexual orientation (oral sex frequency, acts of sexual variety, and communication, etc.).	
Frederick, 2021 [26]	Sexual satisfaction (1–7 scale); own orgasm frequency (0–4 scale); sex frequency; and orgasms in the past month	Sexual satisfaction—mean (sd): HSW 4.5 (1.8), LW 4.6 (1.7); own orgasm frequency—mean (sd): HSW 2.72 (1.23), LW 3.29 (1.04); sex frequency—“more that once a week”: HSW 69% LW 52%; orgasms in past month—“usually always”: HSW 66% LW 85%; and orgasm frequency “never”: HSW 18% LW 8%;	Compared to HSW, LW engaged in more behaviors tied to intimacy and emotional connection. They gave and received oral sex more often, had sex for longer periods of time, and experienced orgasms more routinely. LW were also more likely to engage in manual stimulation of genitals and use sex toys.	
Garcia, 2014 [27]	Percentage of sex that includes orgasm	Percentage of sex that includes orgasm—HSW 75%, L 90 %, B 72%; frequency of orgasm—“never”: HSW 7.5%; LW 2.2%; and frequency of orgasm—“usually always”: 51.2%, 71.5%	In their rates of orgasm occurrence, LW had higher average mean occurrence rates than HSW did. LW reported higher mean orgasm occurrence rates and higher intraindividual variation than HSW did; that is, LW responses were more widely distributed than those among HSW.	
Holt, 2020 [28]	Frequency of sexual activity (past 12 months); orgasm frequency; arousal difficulties; and sexual problems	Frequency of sexual activity—more that once a week: HSW 55.6% LW 46.6%; orgasm with partner—“most of the time”: HSW 68.4% LW 79.4%; orgasms—“never”: HSW 5.2% LW 2.9%; arousal difficulties—HSW 15.8% LW 8.3%; sexual problems—“not at all”: HSW 47.7% L 57.4%	HSW may be more likely to use the role of sexual activity in the relationship as a key barometer for the sexual health of the relationship; LS thought the use of sex materials and experiences with additional partners were more important than HSW did. Minority sexual identities may reflect an associated comfort with sex positivity and disregard of more traditional, sex-negative values.	
Joyner, 2020 [29]	Satisfaction sexual relation (0 to 1)	Satisfaction sexual relation—mean (sd): HSW 0.82 (0.01) LW 0.80 (0.03)	Before and after controlling for a rich set of variables, male and female respondents in same-sex relationships failed to differ from their counterparts in different-sex relationships in their levels of commitment, satisfaction, and emotional intimacy.	
Moreno-Domínguez, 2019 [30]	Frequency of sexual activity	Frequency of sexual activity—more that once a week: HSW 65.9% LW 68.3%	No sexual-orientation-based differences were found for frequency of sexual activity, relationship status, or sexual dissatisfaction. However, body dissatisfaction did exert a lesser influence on sexual dissatisfaction in LW compared to HSW.	
Peixoto, 2015 [31]	Orgasmic difficulties, lack of sexual desire, arousal difficulties, and sexual pain	Experienced difficulties in reaching orgasm—HSW 11.5% LW 6.7%; lack of sexual desire—HSW 9.8% LW 6.7%; arousal difficulties—HSW 8.9% LW 5.6%; and experienced sexual pain—HSW 13.3% L 9.8%	Findings suggested specificities in frequency of self-perceived sexual problems, according to sexual orientation. Overall, findings indicated that HSW reported more sexual problems than LW did. Current data suggest that LW reported fewer difficulties in reaching orgasm.	

## Data Availability

All data included in the review are available under request.

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
