# Peer review of "Sexual Satisfaction among Lesbian and Heterosexual Cisgender Women: A Systematic Review and Meta-Analysis"

_healthcare, 2023, doi:10.3390/healthcare11121680_

Round 1

Reviewer 1 Report

Thank you for the opportunity to review “Sexual Satisfaction among Lesbian and Heterosexual Cisgender Women: A Systematic Review and Meta-Analysis”, an interesting topic that will add greatly to the literature around sexual wellbeing.  I feel the authors do a good job with this review, and after consideration of my minor suggestions and comments below, I feel this publication is ready for print.

Minor points:

Shouldn’t the author affiliations have city, country?

Line 19 (abstract): typo? “OR =1.98 (OR = 1.73, 2.27)”, should the second OR be 95% CI or some other indicator of uncertainty in your estimates?

Line 24: reaching should be reached?

Line 102: “non” should probably be “not”

Line 138: what do you mean by “college-frequency”?  Does this mean “had attended some college”?  I’m not familiar with the phrase.

Figure 1 has some formatting issues to address before publication

Table titles: I recommend a more descriptive title for your tables.  I write mine so that were it to fall on the ground and be separated from the draft, someone could pick it up and understand exactly what it was for.  This is optional, I suppose, but something for you to consider.

Table 2: formatting and column headers look awkward/illegible

I’d recommend better descriptive titles for your Figures too.  Figures 2+ took me a while to digest and interpret.  What is “M-H” in each of those figures? 

Line 191: referred should be reported

Line 251: typo or unclear.  Please revisit this sentence.

Other comments:

Lines 97+: What do you mean by “(6) settings: community”?

Lines 98+: Could you also please expand on your exclusion criteria 1 and 2.  Did the studies have to be exclusively focused on transgender or nonbinary to be excluded?  Or would you exclude a paper with lesbian, heterosexual, and transgender/nonbinary if all three groups would included?   Please also explain what you meant by “global sexual minority outcomes”.   I’m struggling to understand what that might be.

Line 126: What is I2?

Line 129: sensitivity analysis – just a point of clarity, you ran each analysis again, excluding each individual study in turn, to see how one study might affect the results of your overall meta analysis, right?  That’s how I’m interpreting this.

Lines 181+: though technically your meta analysis shows that arousal difficulties is “not significant”, I might recommend that you note the borderline nature of the results.  You repeat this in lines 262+ (discussion).  I think it merits being a little more flexible here, as it’s awfully close to significant.  I will defer to your final judgement, however, as one could go either way here. 

Lines 224: this is a long sentence, and feels like it’s missing a clause or is otherwise incomplete.  Those elements after the “and” are also determinates of sexual satisfaction?

Line 249: some readers may have trouble interpreting ORs.  Perhaps rephase this in words, something to the affect that “LW had significantly less sex than HSW” or “The odds of LW reporting frequent sex were about 57% that of HRW” or whatever you feel is appropriate and clear.

Minor typos, noted in my review.  I recommend a final review of the manuscript to catch anything I might have missed, but in general the English is good.

Reviewer 2 Report

38-39 Great rational for the study.

 Sentence in lines 51-53 should be reworded for clarity.

76 - you said "described above" but showed it below - should say "below"

93 - needs a space between "type" and "("

102 - should say "not" not "non"

Fig 1 - Great flowchart to explain the exclusion process

Table 2 - The formatting for the first row and the last columns should be edited to prevent the chaotic format.

Table 3, 3rd row, 3rd column - This was repeated, said earlier, same paragraph: "LW had a 7-fold higher risk for premature orgasm than HSW."

Table 3, p. 8 of 14, 1st row, 2nd column - "orgasm" is misspelled.

150 - spacing issue after "major"

157 - floating heading

158 - 5,339 what? participants?

Figure 2 - It is in a different font size, looks fuzzy, should mesh with the article.

Figure 3; 4 - same comment as above for two figures below - formatting and spacing issues

196 - Spacing is problematic here

202 - Spacing is problematic here

221-234 - Since this does not tie into new global findings and only seems to reference prior literature from the 11 studies, why isn't it in the lit review? Why is it mentioned in the conclusion? It does not seem to fit here.

235 - spacing issue

251-254 - This is a big issue, because if sexual intercourse is not being defined properly for both LW and HSW, the difference might just be due to the way they are defined.

Included above.

Reviewer 3 Report

Authors should be congratulated for their work. Sexual satisfaction of women is a complex field that must be better understood and studied. The manuscript is well written, and the tables are quite confusing but my main concern is relative to the outcome "sexual satisfaction" measured.  Specifically, referring to the sentence at page 3, lines 110-115: "Sexual satisfaction was evaluated through various parameters that included the frequency with which the woman reached orgasm in a sexual relationship, the degree of difficulty in becoming aroused, pain during or after sex, lack of interest in sex, frequency of sexual activity and global sexual satisfaction" several aspects must be clarified:

- Was a questionnaire used for the evaluation of the outcome?

- Were There cut-offs for every item of the outcome measured? Were they binary (yes or no) or numerically described by the papers collected?

- How could authors compare different results, if they were written differently??

Round 2

Reviewer 3 Report

Authors answered properly to my questions